# A Workcation Improves Cardiac Parasympathetic Function during Sleep to Decrease Arterial Stiffness in Workers

**DOI:** 10.3390/healthcare10102037

**Published:** 2022-10-15

**Authors:** Hideyuki Negoro, Ryota Kobayashi

**Affiliations:** 1Faculty of Medicine, Nara Medical University, Nara 634-8521, Japan; 2Harvard Center for Polycystic Kidney Disease Research, Boston, MA 02115, USA; 3Faculty of Life & Environmental Sciences, Teikyo University of Science, Tokyo 120-0045, Japan

**Keywords:** workcation, arterial stiffness, autonomic nerve activity, physical activity

## Abstract

A “Workcation” (combining work and vacation) has become increasingly common. Traditionally, the workcation focus has been on productivity; however, data showing associations between workcations and improvements in employees’ health are lacking. Therefore, this study examines the effects of a workcation on blood pressure, arterial stiffness, heart rate, autonomic nervous system function, and physical activity. Twenty healthy employees participating in a five-day workcation project at a large private company agreed to participate in this study. Data on arterial stiffness, heart rate, autonomic nerve activity, and physical activity were collected before, during, and after the workcation. Arterial stiffness, blood pressure, and heart rate significantly decreased (*p* < 0.05); meanwhile, physical activity levels and parasympathetic function during sleep significantly increased during the workcation (*p* < 0.05). Thus, a workcation implies a new way of working that improves employees’ cardiovascular indices and parasympathetic function during sleep.

## 1. Introduction

The COVID-19 pandemic had a major impact on work and home life, changing the daily routine of workers. For example, Mojtahedzadeh et al. suggested that contact restrictions related to the COVID-19 pandemic caused many companies to allow employees to work from home for infection-control reasons [1]. Toniolo-Barrios and Pitt reported that employees who worked at home tended to work longer hours than office employees [2]. James et al. found that increased workloads and irregular working hours caused negative changes in circadian rhythms, negatively affecting autonomic function and sleep [3]. Adverse sleep effects are associated with increased arterial stiffness [4], a major independent risk factor for cardiovascular disease [5,6], which is the number one cause of death in the world [7]. Therefore, social countermeasures to improve markers of cardiovascular health are needed.

Lifestyle modification is the first-line approach to primary and secondary prevention of cardiovascular disease [8]. Currently, the Japanese government is promoting workcations, particularly in the private sector [9]. A workcation is a coined term for “combining work and vacation”, and it has gained general agreement as a new way of working whereby people enjoy a relaxing vacation while working in an unusual environment, such as a resort [9]. However, according to Brigitta Pecsek, the academic literature on workcations is insufficient, and the concept has been criticized for inducing work and leisure instability and increasing personal stress [10].

Indeed, although the concept (program) of a workcation has not been established, it is possible that workcations introduce regularity and stabilize work and leisure time, thereby improving circulatory dynamics through the regulation of autonomic function. Therefore, it is necessary to clarify whether workcations have health benefits from a medical perspective.

The purpose of this study was to determine the effects of workcations on cardiovascular health markers, autonomic function, and physical activity in 20 healthy men and women who took workcations in a resort area with forests and sea far from Tokyo, the Japanese capital. The study’s hypothesis was that a workcation would promote parasympathetic activity, physical activity levels during sleep, and improve cardiovascular health markers.

## 2. Materials and Methods

### 2.1. Participants

Twenty healthy Japanese men and women (age: 33.9 ± 8.9 years; 9 men and 11 women; height: 163.9 ± 8.2 cm; weight: 60.7 ± 11.0 kg; body fat percentage: 26.5 ± 9.3%; body mass index (BMI): 22.3 ± 2.8 kg/m^2^) working in a large Japanese private company were recruited for this study. Individuals with jobs involving telecommuting, nonsmokers, and individuals without abnormalities in blood and urine tests, chest X-rays, and electrocardiography (ECG) in the previous year were considered eligible. The exclusion criteria included severe physical symptoms, presence of a pacemaker or another implantable medical device, and abnormalities in physical examination.

To determine the appropriate sample size, a power analysis was performed using G* Power 3 [11]. In our laboratory, the guaranteed effect size for blood pressure and arterial stiffness is 0.5. Using an analysis of variance, it was determined that 15 participants were needed to detect a difference with 80% power and a 5% two-tailed alpha level. An additional five participants were added to account for dropouts. The study was conducted in accordance with the tenets of the Declaration of Helsinki. Ethical approval was obtained from the Teikyo University of Science Ethics Committee (approval number: 21A009). This study was registered with the University Hospital Medical Information Network Center (study number: UMIN000045968) and conducted in accordance with the guidelines for human experimentation published by the Ethics Committee of the Teikyo University of Science. Verbal and written informed consents were provided by all study participants. The study is registered at the University Hospital Medical Information Network Center (UMIN Center) (UMIN Study No. UMIN000045968).

### 2.2. Study Design

The 20 participants completed a 5-day workshop at the Harvest Hotel, a membership resort hotel in Katsuura City, Chiba Prefecture (10 participants) and Hamamatsu City, Shizuoka Prefecture (10 participants). The two hotels are affiliated five-star resort hotels with ocean views and similar room decor, food, and views. A workcation program was introduced in this study. All participants were required to wake up at 7:00 a.m. to enjoy the sunlight and drink a glass of water and have a uniform breakfast, shower, and a lymphatic massage at 7:30 a.m. From 9:00 to 10:00 a.m., participants exchanged opinions. From 10:00 to 11:30 a.m., they performed important work in 90-min cycles. From 11:30 a.m. to 12:00 p.m., they performed exercises with a steady rhythm, such as walking or radio calisthenics. At 12:00 p.m., the participants took a lunch break and a nap. At 2:00 p.m., they performed work that enhances creativity and memory; and at 4:00 p.m., they finished their workday. From 6:00 to 7:00 p.m., the participants performed exercise, including jogging or strength training. A uniform dinner was served from 8:00to 9:00 p.m., and following that, the use of electronic devices was prohibited; at 10 p.m. was bath time. Only water, herbal tea, and milk were allowed after 11 p.m. At midnight, the participants went to bed with the bedroom lights off and the room temperature adjusted (Table 1).

To assess the effects of the workcation, measurements were taken before, during, and after the intervention. The participants’ arterial stiffness, blood pressure (BP), and heart rate (HR) were measured before breakfast. Autonomic function was measured 24 h. Physical activity levels were measured daily (Figure 1).

### 2.3. Body Composition

The participants’ heights were measured in 0.1-cm increments using a height meter. Bodyweight, body fat percentage, and BMI were measured to the nearest 0.1 kg using a precision instrument body composition analyzer (WB-150 PMA; Tanita, Tokyo, Japan).

### 2.4. Arterial Stiffness

Arterial stiffness was measured using a medical electronic blood pressure monitor, AVE-2000Plus (Shisei Datum Co., Ltd., Tokyo, Japan). Measurements were taken in the sitting position within 1 h of waking, after urination, before meals, and after 5 min of rest. Noninvasive assessments were performed using an oscillometric technique with an oscillometric sensor attached to the left upper arm. Considering that the characteristics of the central blood pressure waveform were reflected in the pulse waveform observed at the brachial cuff, the arterial velocity pulse index (AVI) was calculated and evaluated as an index of systemic arterial stiffness from the ratio of brachial artery dilation. Brachial artery relaxation (cardiac dilatation) degree variability (Vr) and cardiac systolic (cardiac systolic) degree variability (Vf) were presented relative to Vr/Vf [12]. If the artery is soft when the pulse wave from the heart reaches the brachial artery, the artery is greatly dilated; if the artery is stiff, its ability to dilate is limited. In this case, the amplitude reached a plateau near the point where it originally reached its maximum, and the pulse wave amplitude became trapezoidal. From this pulse wave amplitude graph, cuff pressure and arterial volume were calculated to determine the arterial pressure-volume index (API), an indicator of peripheral arterial stiffness [12]. Daily coefficients of variation (CV) were 2.9 ± 0.5% and 3.6 ± 1.1% for AVI and API, respectively. Measurements were taken three times in a row, and the average of the two closest values was calculated.

### 2.5. BP and HR

Brachial artery systolic BP (SBP), mean arterial pressure (MAP), diastolic BP (DBP), pulse pressure (PP), and HR were assessed noninvasively by recording blood pressure waveforms using an oscillometric method. The sensor was attached to the left upper arm in a sitting position using a medical electronic blood pressure monitor, the AVE-2000Plus (Shisei Datum Co., Ltd., Tokyo, Japan). The same device used to measure arterial stiffness was used to simultaneously measure BP and HR [12]. The estimated aortic blood pressure was calculated using a medical electronic blood pressure monitor (AVE-2000Plus, Shisei Datum Co., Ltd., Tokyo, Japan) using the following formula: estimated aortic SBP = 0.1152 × age + 0.7512 × brachial plexus SBP + 0.3095 × brachial plexus DBP + 0.1884 × AVI + 0.4001 × API − 0.1105.

Measurements were taken three times in a row, and the average of the two closest values was calculated. The daily coefficients of variation in our laboratory were 2 ± 1%, 2 ± 2%, and 2 ± 1% for brachial BP, aortic SBP, and HR, respectively.

### 2.6. Autonomic Nerve Activity

Heart rate variability was measured noninvasively using a wearable HR sensor WHS-1/RRD-1 (Union Tool Co., Ltd. Tokyo, Japan). The R–R interval (RRI) is the interval between the R and R waves that appear on the ECG. Since the amplitude of the R waves is larger than other ECG components (P, Q, S, and T), the power spectrum analysis of the frequency components of the RRI time series can be used to obtain the low-frequency (LF) component of power from 0.04 Hz to 0.15 Hz, which is influenced by the sympathetic and parasympathetic nervous systems, and the high-frequency (HF) component of power from 0.15 Hz to 0.4 Hz, which is influenced by the parasympathetic nervous system. Many clinical and experimental studies have demonstrated a strong association between LF power and sympathetic activity and, HF power and cardiac parasympathetic activity [13,14]. Taking the ratio of LF to HF norm was used to assess the balance between sympathetic and parasympathetic nervous systems [15].

### 2.7. Physical Activity

The amount of physical activity was measured using a 3-axis accelerometer (HJA-750C; Active Style Pro, Omron, Kyoto, Japan). All participants wore a triaxial accelerometer on their waist at all times except during sleep and baths. Data on daily walking calories, daily living activity calories, total calories, walking activities, daily living activities, total activities, number of steps, walking time, and activity intensity (low, medium, and high) were recorded [16].

### 2.8. Statistical Analysis

Data are presented as mean ± standard deviation (SD). The normality of the data and the homogeneity of variance were examined using the Shapiro–Wilk and Levene’s tests, respectively. The change in each measurement before, during, and after the workcation is presented as mean and 95% confidence interval for each group. Parametric analysis was performed for the measurement items using one-way analysis of variance with repeated measures. When the assumption of sphericity was violated (Mauchly’s test), the analysis was adjusted using the Greenhouse–Geisser correction. Post-tests were conducted using the Bonferroni method to identify the changes in each intervention. Correlations among AVI, API, brachial SBP, aortic SBP, energy consumption of daily life activities, high-intensity life activities, and HF norm during sleep were examined using the Pearson product-moment correlation coefficient. A statistical analysis was performed using IBM SPSS Statistics for Macintosh, version 25.0 (IBM Corp., Armonk, NY, USA). The statistical significance was set at *p* < 0.05, and all *p*-values were two-sided. 

## 3. Results

### 3.1. Arterial Stiffness

Figure 2 shows the changes in API and AVI. The API decreased significantly on days 2 (*p* < 0.05), 3 (*p* < 0.01), and 4 (*p* < 0.05) of the workcation relative to pre-workcation values. There was no difference between pre- and post-workcation values. The AVI decreased significantly on days 2 (*p* < 0.05), 3 (*p* < 0.05), and 4 (*p* < 0.05) of the workcation relative to pre-workcation values. There was no difference between pre- and post-workcation values or in measurements pre- and post-workcation.

### 3.2. BP, HR, and Double Product (DP)

Figure 3 shows BP and HR values. Brachial SBP, DBP, and MAP decreased significantly on days 2 (*p* < 0.05, *p* < 0.05, *p* < 0.01, respectively) and 4 (*p* < 0.01, *p* < 0.05, *p* < 0.01, respectively) of the workcation relative to pre-workcation levels, with no significant differences between pre- and post-workcation levels. Aortic SBP decreased significantly on days 2 (*p* < 0.01), 3 (*p* < 0.05), and 4 (*p* < 0.01) relative to pre-workcation levels, with no significant differences between pre- and post-workcation values. The HR decreased significantly on day 2 (*p* < 0.05) relative to the pre-workcation value, with no significant differences between pre- and post-workcation values. Brachial PP in the upper arm remained unchanged before and after the workcation. DP significantly decreased on days 2 (*p* < 0.01), 3 (*p* < 0.01) and 4 (*p* < 0.01) relative to pre-workcation levels, with no significant differences between pre- and post-workcation values.

### 3.3. Autonomic Nerve Activity

Figure 4 shows the average autonomic function measurements for 24 h and while awake and asleep. The HF norm component during sleep increased significantly (*p* < 0.05) during the workcation relative to that before the workcation. However, there was no change in LF/HF and LF norm over 24 h, LF/HF and LF norm while awake and asleep, HF norm over 24 h, and HF norm while awake. 

Table 2 lists the changes in physical activity. Calories consumed for daily living activities significantly increased during the workcation relative to pre-workcation levels (*p* < 0.01); however, calories consumed for exercise did not change significantly between these two periods. In terms of exercise intensity, high-intensity activities significantly increased during the workcation period relative to those in the pre-workcation period (*p* < 0.01); however, low- and moderate-intensity activities did not change significantly between the two periods. The number of steps and walking time did not change significantly during the workcation period compared with those in the pre-workcation period.

### 3.4. Correlations between Arterial Stiffness, Sbp, Energy Consumption of Daily Life Activities, High-Intensity Life Activities, and Hf Norm during Sleep

Figure 5 shows the correlations between AVI, API, brachial SBP, aortic SBP, energy consumption of daily life activities, high-intensity life activities, and HF norm during sleep. The HF norm during sleep was negatively correlated with API (*p* < 0.01, r = 0.53) and energy consumption of daily life activities (*p* < 0.01, r = 0.75). There was no correlation between the HF norm during sleep and AVI, brachial SBP, aortic SBP, and high-intensity life activities.

## 4. Discussion

The key findings of this study were the increased HF norm component during sleep and decreased API and AVI during the workcation. Moreover, brachial and aortic SBP and DP decreased during the workcation. A correlation was found between the HF norm component during sleep and API during the workcation. These findings may contribute to the prevention of future atherosclerosis in employees who have some problems with sleep.

To the best of our knowledge, this is the first study to investigate the effects of workcations on cardiovascular health markers. Studies on the regularity of life have shown beneficial effects on sleep quality [17]. Sleep duration in Japan is short by global standards, and approximately 60% of the population has some kind of sleep problem [18]. In general, the parasympathetic nervous system becomes dominant during nocturnal sleep, and the HR decreases as deep sleep is achieved and increases upon awakening [19]. Indeed, HR variability during sleep has been reported to predict sleep function [20]. According to Busek, the HF norm is higher in non-REM sleep than in REM sleep in normal subjects [21]. It has also been reported that individuals with poor sleep quality have lower HF norm than those with good sleep quality [22]. Furthermore, the HF power has been reported to be strongly associated with parasympathetic activity [23,24,25,26,27,28,29]. In the present study, the HF norm during sleep was higher during the workcation than before the workcation. In other words, the increase in the HF norm during sleep may have been due to increased parasympathetic activity. Therefore, elevation of the HF norm during sleep during the workcation in the present study may have been involved in the improvement of sleep quality. However, this study did not measure sleep quality in detail by electroencephalography (EEG) or other methods; therefore, further investigations are warranted.

In this study, powerful indices of vascular function were selected and utilized to follow potential changes induced by the workcation. The API and AVI were evaluated as indices of arterial stiffness. The API reflects the stiffness of the peripheral arterial wall; meanwhile, the AVI reflects the stiffness of the aorta and the peripheral arterial wall. The workcation intervention decreased the API and AVI, and the decrease persisted until the 4th day of the intervention. Therefore, lifestyle changes associated with a workcation may decrease arterial stiffness. In fact, arterial stiffness can be affected by mental stress [25], toxins, and other factors [26] as well as various environmental factors. In the present study, the HF norm during sleep increased during the intervention, and a correlation was found between the HF norm during sleep and API. It has been reported that most vascular functions can be modulated by the contractile state of smooth muscle cells in the arterial wall, and that autonomic function may be involved in changes in vascular function [27]. Thus, increased parasympathetic activity during sleep may be involved in lowering the arterial stiffness during the workcation in the present study. However, we found no correlation between the HF norm during sleep and the AVI during the intervention. However, several studies have reported an association between autonomic function and aortic stiffness in healthy subjects. Specifically, prior research indicates that changes in HF norm may not be involved in aortic stiffness in healthy subjects [30]. Furthermore, another study examining the effects of autonomic function on arterial stiffness reported that in healthy participants, the autonomic function played no direct role in regulating aortic stiffness [31]. Consistent with these results, the findings in our study revealed no relationship between changes in HF norm and AVI during the intervention. This may involve organic influences, such as the greater amount of elastic tissue and lesser smooth muscle in central arteries than in peripheral arteries, which accommodate the high pressures caused by ventricular ejection [32]. Future studies should examine the pulse wave velocity between the carotid and femoral arteries (an indicator of aortic stiffness) and between the femoral and ankle arteries (an indicator of peripheral arterial stiffness) to determine their relationship with cardiac autonomic function. Adrenaline, vascular disorders, and sleep quality may be related. In healthy adults, acute nocturnal aircraft-noise exposure impairs endothelial function in a dose-dependent manner, stimulates adrenaline release, and worsens sleep quality [33]. Conversely, it is possible that adrenergic hormones may be involved in the HF norm improvement and the API decrease during the workcation. Unfortunately, hormones were not measured in this study. Future studies should investigate this issue.

Physical inactivity is the fourth leading cause of all deaths worldwide [34]. Individuals who are continuously inactive have an approximately 27% higher risk of cardiovascular disease; meanwhile, those who increase their level of physical activity decrease their risk of cardiovascular disease by 11% [35]. Physical activity improves circulation, increases energy levels, and improves quality of life [36]. In the present study, energy expenditure by daily activities increased during the workcation intervention. At first glance, the increase in energy expenditure by daily activities during the intervention appears to be slight. However, previous studies have shown that the daily accumulation of a small surplus of energy, approximately 15–60 kcal per day, is a major cause of lifestyle-related diseases [37]. Therefore, it is also important to increase activity and daily energy expenditure through the unusual environment of a workcation in order to reduce future mortality from cardiovascular and other diseases. Physical activity also helps improve sleep quality [38]. In this study, energy expenditure by daily activities increased during the workcation; meanwhile, the HF norm increased during sleep. In addition, a correlation was found between the HF norm during sleep and energy expenditure through daily living activities during the workcation. These findings suggest that the increase in physical activity due to the improved lifestyle rhythm caused by the workcation may have improved sleep quality. It also suggests that the BP decreases in proportion to increased physical activities [39]. In the present study, the brachial artery and central artery SBP decreased during the workcation. Thus, a workcation may decrease BP through increased physical activity.

This study has several limitations. First, cardiovascular health marker levels after the intervention were similar to those before the intervention. This is because the participants returned to their regular lifestyle after the workcation ended. Therefore, there is a need to consider ways of ensuring that participants sustain a similar lifestyle after a workcation. Second, the participants were healthy volunteers from a single firm, so the results cannot be generalized to employees with pre-existing conditions, or employees from other firms. Third, the sample size was small (*n* = 20). Fourth, measurements of hormones such as adrenaline and EEG, which may have important effects on sleep quality and arterial stiffness, were not performed. Future research should include employees of different companies and stratify participants by age, sex, disease, and occupation to increase the credibility of the effects of workcations. Finally, there is no consensus regarding the protocol of the workcation and health benefits, which will need to be detailed in future studies.

## 5. Conclusions

The workcation improved vascular function in a positive manner and increased parasympathetic tone during sleep. In addition, physical activity increased while BP and cardiac load decreased. These results indicate that increased parasympathetic activity during sleep may have favorable effects on cardiovascular markers. Although this study does not support the concept of a workcation, it provides clinically meaningful findings in terms of the health benefits of a workcation. We believe that this paper will be of interest to the readership of your journal because these findings have implications for the long-term health of employees at a time when workplace norms are shifting, and when home working during the COVID-19 pandemic may lead to a deterioration in physical and mental health.

## Figures and Tables

**Figure 1 healthcare-10-02037-f001:**
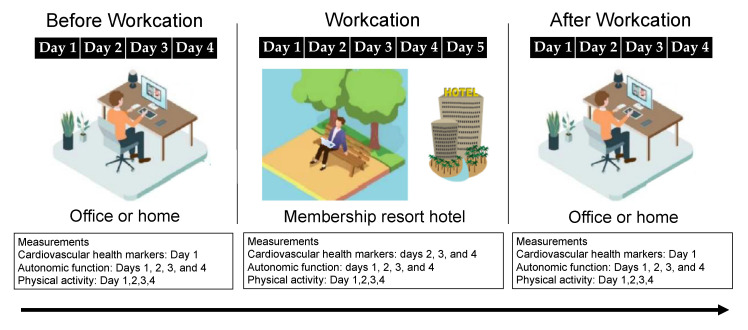
Study design.

**Figure 2 healthcare-10-02037-f002:**
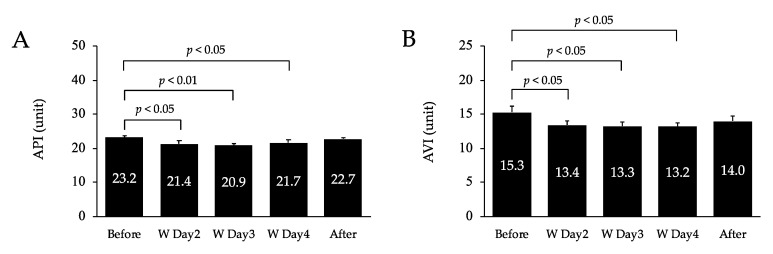
Changes in peripheral arterial stiffness (**A**) and systemic arterial stiffness (**B**) before, during, and after the workcation. Values are mean ± SD. W, workcation. API, arterial pressure-volume index; AVI, arterial velocity pulse index.

**Figure 3 healthcare-10-02037-f003:**
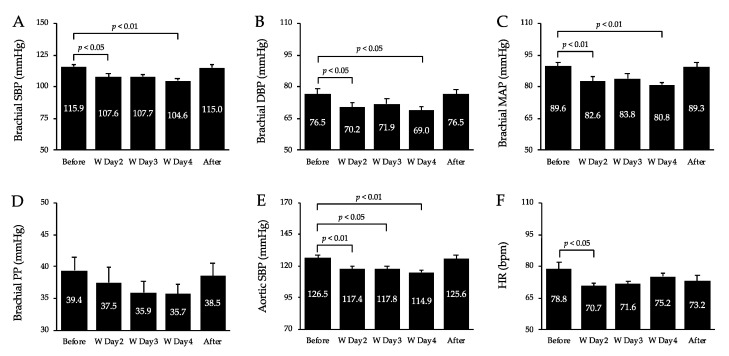
Changes in brachial SBP (**A**), DBP (**B**), MAP (**C**), PP (**D**), aortic SBP (**E**) and HR (**F**) before, during, and after the workcation. Values are mean ± SD. W, workcation; SBP, systolic blood pressure; DBP, diastolic blood pressure; MAP, mean arterial pressure; PP, pulse pressure; HR, heart rate.

**Figure 4 healthcare-10-02037-f004:**
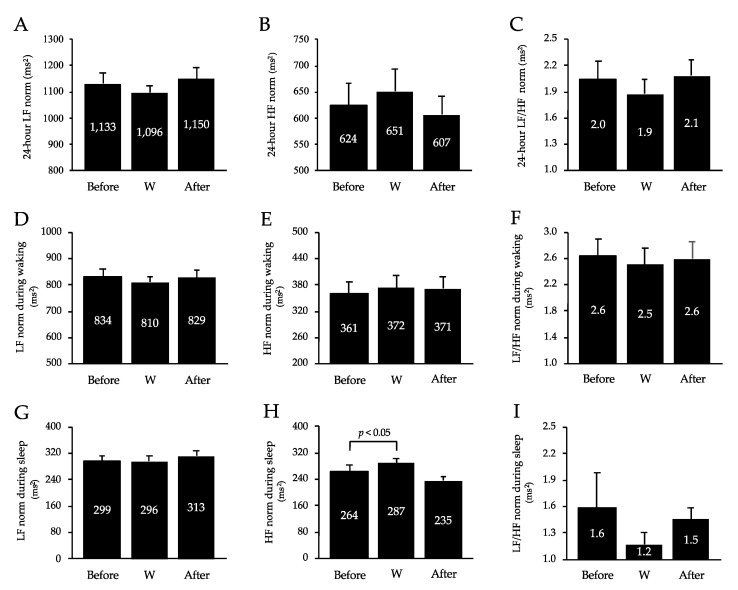
Changes in 24 -hour LF norm (**A**), HF norm (**B**), and LF/HF norm (**C**), LF norm (**D**), HF (**E**), and LF/HF (**F**) during waking, and LF norm (**G**), HF (**H**), and LF/HF (**I**) during sleep. Values are mean ± SD. W, workcation; LF, low-frequency; HF, high-frequency.

**Figure 5 healthcare-10-02037-f005:**
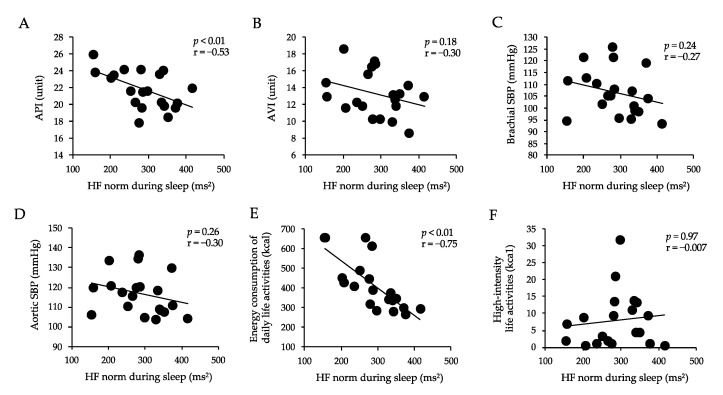
Correlations between API (**A**), AVI (**B**), brachial SBP (**C**), aortic SBP (**D**), energy consumption of daily life activities (**E**), high-intensity life activities (**F**), and HF norm during sleep. HF, high-frequency; API, arterial pressure-volume index; AVI, arterial velocity pulse index; SBP, systolic blood pressure. Values are mean ± SD.

**Table 1 healthcare-10-02037-t001:** Workcation program.

Time	Activity
7:00 am	Wake up time.
	Participants drank a glass of water.
7:00 am–8:00 am	Participants had breakfast within an hour of waking up.
	Participants ate breakfast, especially foods that contained carbohydrates (rice, bread, etc.) and proteins (meat, fish, etc.), and took a hot shower or had a lymphatic massage to expel waste.
9:00 am–10:00 am	Participants performed intellectual work that required calm judgment.
10:00 am–11:30 am	Participants made important decisions and performed tasks that did not allow for errors in judgment (intensive work was completed in 90-min cycles.)
11:30 am	Participants performed exercises with a steady rhythm, such as walking or radio calisthenics.
12:00 pm	Lunch at a specific time. After lunch, nap time for no more than 15 min.
2:00 pm	Participants performed tasks that required creativity, such as planning, and tasks that required a good memory.
4:00 pm	Discussions, information exchanges, paperwork, and other calm tasks to finish up the day.
6:00 pm–7:00 pm	Exercise, such as strength training and walking.
8:00 pm	Participants ate a moderate number of calories and vegetables for dinner.
9:00 pm	Participants turned off cell phones, smart phones, and computers.
10:00 pm	Half-body bath time at a lukewarm temperature (38–41 °C).
11:00 pm	Only water, herbal tea, or milk before going to bed.
12:00 am	Bedtime.

**Table 2 healthcare-10-02037-t002:** Physical activity.

Variable	Before	W	After
Energy consumption of walking (kcal)	241.2 ± 13.5	255.3 ± 22.2	263.9 ± 18.6
Energy consumption of daily life activities (kca1)	319.1 ± 21.3	381.1 ± 25.5 *	326.7 ± 34.6
Total energy consumption (kcal)	560.3 ± 28.1	636.3 ± 33.8	590.6 ± 48.9
Walking exercises (EX)	3.7 ± 0.2	3.7 ± 0.5	4.0 ± 0.3
Activities of daily living exercises (EX)	1.1 ± 0.1	1.7 ± 0.1	1.0 ± 0.1
Exercise total (EX)	4.8 ± 0.2	5.4 ± 0.5	5.0 ± 0.4
Walking time (minutes)	77.8 ± 4.7	79.9 ± 5.4	82.7 ± 5.6
Duration of low-intensity activity (minutes)	575.1 ± 16.0	636.3 ± 19.4	588.3 ± 31.6
Duration of medium intensity activity (minutes)	66.5 ± 2.5	65.5 ± 4.2	69.4 ± 4.7
High-intensity activity time (minutes)	3.1 ± 1.0	8.0 ± 2.2 *	3.4 ± 4.7

Values are mean ± SD. W, workcation. * *p* < 0.05 vs. Before.

## Data Availability

The data presented in this study are available on request from the corresponding author.

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
