# Peer review of "A Workcation Improves Cardiac Parasympathetic Function during Sleep to Decrease Arterial Stiffness in Workers"

_healthcare, 2022, doi:10.3390/healthcare10102037_

Round 1
Reviewer 1 Report
This study is focused on effect of ’Workcation’ (combining work and vacation) on cardiovascular health markers which is interesting and innovative.
1. Some spelling error in Figure 1 need to be corrected.
2. The effect of workcation is most likely correlated to hormone changes, such as epinephrine. It is better if the author measure the hormone changes in this study.
Author Response
We appreciate your attention to our manuscript. We feel that our submission has been considerably enhanced by implementing your valuable advice. We addressed the specific comments as outlined in detail below, and modifications to the revised text are highlighted in red. We hope you will agree that the changes have rendered our report acceptable for publication.
1. Some spelling error in Figure 1 need to be corrected.
We have corrected figure 1.
2. The effect of workcation is most likely correlated to hormone changes, such as epinephrine. It is better if the author measure the hormone changes in this study.
Page 8, lines 277-280
We appended Hormonal Influences.
Author Response
We appreciate your attention to our manuscript. We feel that our submission has been considerably enhanced by implementing your valuable advice. We addressed the specific comments as outlined in detail below, and modifications to the revised text are highlighted in red. We hope you will agree that the changes have rendered our report acceptable for publication.
Dear Editors, Dear Authors,
Congratulations to the Authors on the preparation of this publication. The discussed topic is very interesting, innovative and relevant to public health. Employee health should be a priority for every employer to keep him productive and working well. Working conditions are often not conducive to this. Therefore, the discussed topic is important, however, it is difficult to assess for example 20 people from one company. The study was laborious for the authors, I respect the contribution of the work, but it is difficult to draw conclusions on the basis of such "limited" studies. I propose to extend the research and then re-present the publications.
Page 9, lines 301-312
We have added a note on limitations and limits of the study. We will be working on details to resolve the limitations of the study.
Reviewer 3 Report
The authors have conducted a pre-post study to determine the effects of a combination of work and vacation (workcation) on arterial stiffness, blood pressure and heart rate, among others on workers of a private company. It is necessary to make some clarifications before publishing.
1. Methods: How was the sample calculated?
2. Methods: What criteria were used to form A and B groups? What is the rationale for this? Why are A and B groups not compared throughout the study?
3. Results: Are there differences between women and men?
4. Design: Is the "workcation" term appropiate for this study?
Author Response
We appreciate your attention to our manuscript. We feel that our submission has been considerably enhanced by implementing your valuable advice. We addressed the specific comments as outlined in detail below, and modifications to the revised text are highlighted in red. We hope you will agree that the changes have rendered our report acceptable for publication.
The authors have conducted a pre-post study to determine the effects of a combination of work and vacation (workcation) on arterial stiffness, blood pressure and heart rate, among others on workers of a private company. It is necessary to make some clarifications before publishing.
1. Methods: How was the sample calculated?
Page 2, lines 64-68
We used G-power for our sample size calculations.
2. Methods: What criteria were used to form A and B groups? What is the rationale for this? Why are A and B groups not compared throughout the study?
Page 2, lines 78-81
The hotels where they stay are different from each other and are represented as a group. However, since the hotels were affiliated and had the same rooms and environment, we cannot think of any element of comparison.
Therefore, we have revised the text.
3. Results: Are there differences between women and men?
There was no difference.
It has been decided to study the details of the gender difference with a larger number of participants in the future.
4. Design: Is the "workcation" term appropiate for this study?
According to the Japanese government, workcation are a way to work away from the office while also taking vacations, and are characterized by the ability to proceed with work in a travel destination, such as a resort or a rural area.
In this study, employees lived in resorts, combining vacation time while proceeding with work.Therefore, we consider this to be a work vacation.However, since there is no consensus on the content of work vacations that can improve health, future research should clarify the details.
Round 2
Reviewer 2 Report
letter attached

Author Response
We appreciate your attention to our manuscript. We feel that our submission has been considerably enhanced by implementing your valuable advice.
We will publish the article in its current format.
Reviewer 3 Report
The authors did a good job responding to my previous concerns.
Author Response
We appreciate your attention to our manuscript. We feel that our submission has been considerably enhanced by implementing your valuable advice.